# Low husband involvement in maternal and child health services and intimate partner violence increases the odds of postpartum depression in northwest Ethiopia: A community-based study

Azmeraw Ambachew Kebede[1], Dereje Nibret Gessesse[1], Mastewal Belayneh Aklil[1], Wubedle Zelalem Temesgan[1], Marta Yimam Abegaz[1], Tazeb Alemu Anteneh[1], Nebiyu Solomon Tibebu[1], Haymanot Nigatu Alemu[1], Tsion Tadesse Haile[1], Asmra Tesfahun Seyoum[1], Agumas Eskezia Tiguh[1], Ayenew Engida Yismaw[1], Muhabaw Shumye Mihret[1], Goshu Nenko[2], Kindu Yinges Wondie[1], Birhan Tsegaw Taye[3], Nuhamin Tesfa Tsega[4]*

1 Department of Clinical Midwifery, School of Midwifery, College of Medicine and Health Sciences, University of Gondar, Gondar, Ethiopia, 2 Department of Psychiatry, College of Medicine and Health Sciences, University of Gondar, Gondar, Ethiopia, 3 Department of Midwifery, College of Health Sciences, Debre Berhan University, Debre Berhan, Ethiopia, 4 Department of Women's and Family Health, School of Midwifery, College of Medicine and Health Sciences, University of Gondar, Gondar, Ethiopia

* nuha26tesfa@gmail.com

## Abstract

### Background

Depression is the most common mental health problem that affects women during pregnancy and after child-birth. Postpartum depression, in particular, has both short and long-term effects on the lives of mothers and children. Women's health is a current global concern, but postpartum depression is a neglected issue in the maternal continuum of care and is rarely addressed. Therefore, this study aimed to assess postpartum depression and associated factors in Gondar city, northwest Ethiopia.

### Methods

A community-based cross-sectional study was conducted from August 1st to 30th, 2021 in Gondar city. A cluster sampling technique was employed to select 794 postpartum women. Data were entered by EPI DATA version 4.6 and exported to SPSS version 25 for further analysis. The multivariable logistic regression analysis was carried out to identify factors associated with postpartum depression. The adjusted odds ratio with its 95% confidence interval at a p-value of $\leq 0.05$ was used to declare the level of significance.

### Results

A total of 794 women were included in the analysis, giving a response rate of 98.5%. The prevalence of postpartum depression was 17.25% (95% CI: 14.5, 20.2). Younger maternal

**Data Availability Statement:** All relevant data are within the paper and its Supporting Information files.

**Funding:** The author(s) received no specific funding for this work.

**Competing interests:** The authors have declared that no competing interests exist.

**Abbreviations:** AOR, Adjusted Odds Ratio; ANC, Antenatal Care; CI, Confidence Interval; COR, Crude Odds Ratio; IPV, Intimate Partner Violence; MNCH, Maternal, Neonatal, and Child Health; PNC, Postnatal Care; PPD, Postpartum Depression; SPSS, Statistical Package for Social Science.

age (AOR = 2.72, 95% CI: 1.23, 5.85), low average monthly income (AOR = 2.71, 95% CI: 1.24, 5.91), low decision-making power (AOR = 2.04, 95%CI: 1.31, 3.18), low husband/partner involvement in MNCH care service (AOR = 2.34, 95%CI: 1.44, 3.81), unplanned pregnancy (AOR = 3.16 95% CI: 1.77, 5.62), and experience of intimate partner violence (AOR = 3.13; 95% CI: 1.96, 4.99) were significantly associated with increased odds of postpartum depression.

## Conclusion

In this study, nearly 1/5th of the study participants had postpartum depression. Thus, it is important to integrate maternal mental health services with the existing maternal health care services. It is also crucial to advocate the need for husband's involvement in MNCH care services and ensure women's decision-making power in the household. Moreover, community-based sexual and reproductive health education would be better to reduce risk factors of postpartum depression.

## Introduction

Depression is the most common psychiatric condition that occurs during pregnancy and the postpartum period [1]. Postpartum Depression (PPD) is defined as depressive symptoms such as reduced mood, loss of enjoyment, diminished energy and activity, functional impairment, low self-esteem, and suicidal thoughts or acts that occur within the first year after childbirth [2–4]. The transition to motherhood is considered to be a difficult and emotional transition with significant changes in psychological, social and physiological aspects, and increased susceptibility to mental illnesses including PPD [5]. It is a major public health concern because it affects mothers and newborns, as well as family members and the society at large [6]. In the same way, intimate partner violence (IPV) is a global public health problem within an intimate relationship that causes physical, psychological, or sexual harm [7]. It can cause extensive mental health consequences including depression among victims [8,9]. In Ethiopia, the prevalence of IPV among pregnant women was 28.74 [10] and in central Ethiopia, it was 31.4% postpartum IPV [11].

Around 10%-20% of mothers suffer from depressive symptoms after childbirth worldwide [12]. In sub-Saharan Africa the magnitude of postpartum depression is 18.6% [13]. A meta-analysis study conducted in Ethiopia also showed that 21.5% of women develop postpartum depression [14].

Empirical evidence has found that postpartum depression is linked to impaired mother-infant bonding, child abuse, child neglect, maternal substance abuse, and self-harm [15,16]. In addition, maternal depression has also been linked to poor weight gain, impaired cognitive and motor development in infants, and early discontinuation of breastfeeding because of reduced breast milk production [15,17]. Moreover, maternal depression affects the nutrition of the women that could lead to some morbid conditions like anemia, malnutrition, and hypertension [1,12]. The family can be affected through neglect of family duties and financial strain due to the treatment costs for PPD and low productivity at work [12]. This maternal depression can also lead to suicide, which is a leading cause of death in the first postnatal year, accounting for around 22% of maternal deaths. About 10% of maternal suicide, in particular, is resulted from postpartum mental health problems [18].

Endless crying of babies, painful and cracked nipples, painful delivery wounds, inadequate breast milk, family demands, sleepless nights, and constant fatigue have been associated with postpartum depression [19]. Even though PPD has no single cause, some of the factors are being a first-time mother, history of previous depression [20], domestic violence [21,22], history of substance use, poor social support [22,23], and unplanned pregnancy [20,21].

The 2015 World Health Organization recommendation on measures to promote maternal and child health outcomes during pregnancy, childbirth and the postpartum period include effective implementation of male involvement in the maternal continuum of care [24]. This is because low husband/partner involvement in Maternal, Neonatal, and Child Health (MNCH) care services during pregnancy and the postpartum period is a leading cause of poor maternal health, including PPD [25].

However, the PPD screening tool (i.e. Patient health questionnaire-9) is not incorporated into modern postnatal care guidelines in Ethiopia. This study can help reduce maternal PPD and influencing factors and provide information on strategies targeting on maternal and child health. Therefore, this community based cross-sectional study assessed postnatal depression and associated factors among women who gave birth in the last one year in Gondar City, Northwest Ethiopia.

## Method and materials

### Study design and period

A community-based cross-sectional study was conducted in Gondar city from August 1$^{st}$ to August 30$^{th}$, 2021.

### Study area

Gondar city is found in Amhara national regional state, Central Gondar Zone. It is located 166 km from Bahir Dar (the capital city of Amhara regional state) and 750 km Northwest of Addis Ababa (the capital city of Ethiopia). According to the population projection of Ethiopia, the estimated total population of the city was 432,191, of whom, 224,508 are females. From this, about 133, 477 (30.88%) of females are in the reproductive age (women aged from 15–49 years old) (unpublished data by Amhara regional state, 2021). There are 1 governmental comprehensive specialized referral hospital, 8 governmental health centers, 22 health posts, 1 private primary hospital, and 1 general hospital serving the town.

### Study population and eligibility criteria

The study population included all women who gave birth in the last year (from August 2020 to August 2021) and who resided in the city for at least 6 months in the selected kebeles during the data collection period.

### Sample size determination and sampling procedure

The sample size was determined by using the single population proportion formula by considering the following assumptions: the proportion of postpartum depression 33.82%, which was done in southwest Ethiopia [26], level of confidence 95%, and margin of error 5%. Therefore, the sample size (n) $= \frac{(Z\alpha/2)^2 p(1-p)}{d^2} = \frac{(1.96)^{2*} 0.3382(1-0.3382)}{(0.05)2} = 344$. After considering a design effect of 2 and a non-response rate of 10%, the total sample size was 757. Gondar city has 22 kebeles (the smallest administrative unit) and six kebeles (25% of the total kebeles) were randomly selected by a lottery method. All eligible women in the selected clusters were interviewed. Finally, due to the nature of cluster sampling, 806 women were included in our study.

## Variables of the study

Postpartum depression was the outcome variable whereas maternal age, religion, marital status, mother's educational status, women's occupation, average monthly income, mother's educational status, husband educational status, husband occupation, parity, having ANC visit, place of delivery, mode of delivery, PNC visit, planned pregnancy, intimate partner violence (IPV), decision-making power, social support, and husband/partner involvement in MNCH, family history of mental health problem, known psychiatric illness, having information about mental health during pregnancy, having medical illness and experienced a death of family or friends were independent variable of the study.

## Operational definitions and measurements

**Postpartum depression:** Women who were interviewed and scored ten and above by using patient health questionnaire-9 (PHQ-9) were considered as depressed [27].

**Social support:** The Oslo Social Support Scale (OSS-3) scores ranged from 3–14 with a score of 3–8, poor support; 9–11, moderate support; and 12–14, strong support [28].

**Household decision-making power:** A total of eight questions were prepared to assess the household decision-making power of the women. A score of 2 were given for women who decided independently, 1 for women who decided with their husband, and 0 for decisions made by the husband alone or other person. The minimum and the maximum scores were 0 and 16, respectively. Thus, based on the summative score of variables designed to assess household decision-making power women, who answered above the mean value (8.98) were considered to have higher decision-making power [29].

**Husband/partner involved in MNCH services:** It was composed of nine questions for this study. For each question, the response was given a score of 0 and 1. The total score was 9, with a minimum of 0 and a maximum of 9. Hence, husband involvement with a score above the mean (6.08) was considered as involved [25].

**Intimate partner violence**: Intimate partner is considered as a current spouse, co-habited, current boyfriends, former partner, or spouse. Women were considered to have experienced intimate partner violence, if they said "Yes" to any one of the ranges of sexual, psychological, and physical or any combination of the three coercive acts regardless of the legal status of the relationship with current/former intimate partner, it was considered as IPV [30].

**Having medical illness:** It was defined as a women who presented with at least one of the following medically diagnosed illness: hypertension, diabetes, asthma, cardiac and renal disease [31].

## Data collection tool and quality assurance

The data collection tool was developed by reviewing the literature [8,21–23,26,32–36] and was collected using a structured questionnaire through face-to-face interviews. The questionnaire was prepared in the English version and translated to the local language (Amharic) and back to English to keep uniformity. The questionnaire contains socio-demographic characteristics, obstetric, medical, and maternal health services-related characteristics, social support, husband/partner involvement in MNCH, decision-making power, and intimate partner violence-related questions, and questions assessing PPD. Six BSc and two MSc midwives were recruited for data collection and supervision, respectively. To assure the quality of the data, one day training was given for data collectors and supervisors about the interview technique and supervising the data process. Moreover, pretest was done on 5% of the determined sample size in the Maksegnit district to look for the understandability and appropriateness of the study tool. The completeness of the questionnaire was checked by the supervisors daily.

## Data management and analysis

Data were checked, coded, and entered into EPI Data version 4.6 and exported to SPSS version 25 for further analysis. Descriptive statistics like frequency, mean, and proportion were used to present participants' characteristics. Binary logistic regression was fitted to identify eligible factors and variables having a p-value of $\leq 0.2$ were included in the multivariable logistic regression analysis. In the multivariable logistic regression analysis, a p-value of $\leq 0.05$ with a 95% CI for the adjusted odds ratio was used to claim the level of significance.

## Ethical considerations

BKEthical clearance was obtained from the University of Gondar Institutional Review Board (IRB) (Reference number: V/P/RCS/05/2710/2021). A formal letter of administrative approval was obtained from the selected clusters (kebeles) of Gondar city. Written informed consent was taken from each study participants after a clear explanation of the aim of the study.

# Result

## Socio demographic characteristics of respondents

A total of 806 women were included in the study making a response rate of 98.5%. Among the total study participants, the mean age of the respondents was 29.7 years old (±SD 4.83) and 550 (69.3%) of the respondents were in the age group of 25–34 years. Most (81.6%) of the study participants were Orthodox Christian. Regarding marital status, 718 (90.4%) of study participants were married. More than one quarter, 229 (28.8%) of study participants had completed secondary education. Regarding occupation, 357 (45%) women were housewives, and 457(66.15%) of their husbands were government employees (**Table 1**).

## Obstetrics, medical related, and maternal health service characteristics

Among the total respondents, more than half (56.2%) of women had a parity of two to four. The majority, 773 (97.4%) of study participants had at least one ANC visit whereas 418 (52.6%) of the participants had at least one PNC visit. Six hundred seventy eight (85.4%) and seven hundred thirty seven (92.8%) of the pregnancies were planned and supported by husband/family, respectively. Regarding social support, 228 (28.7%) of the respondents had poor social support. More than half, 416 (52.4%) of women's got their husband/partner's support during maternal, neonatal, and child health care services, and nearly two-thirds, 494 (62.2%) of women had a higher decision making power (**Table 2**).

## Prevalence of postpartum depression and associated factors

The prevalence of postpartum depression among women who gave birth in the last one year was 17.25% (95% CI: 14.5, 20.2). In the binary logistic regression analysis: maternal age, women's educational status, family monthly income, intimate partner violence, women's decision-making power, type of pregnancy, husband/partner involvement in MNCH service, social support, place of delivery, and pregnancy supported by family/partner were found to be a p-value of <0.2 and entered into multivariable analysis. However, maternal age, family monthly income, women's decision-making power, husband/partner involvement in MNCH care service, type of pregnancy, and IPV were significantly associated with postpartum depression in the multivariable analysis.

Women whose age was $\leq 24$ years were 2.72 times more likely to develop postpartum depression compared with women whose age was $\geq 35$ years (AOR = 2.72, 95% CI: 1.23, 5.85). Women whose average monthly income $\leq 1000$ Ethiopian Birr (ETB) were 2.71 times more

**Table 1. Socio-demographic characteristics of women who gave birth in the last one year in Gondar city, North-west Ethiopia, 2021 (n = 794).**

| Variables | Frequency | Percentage (%) |
|---|---|---|
| **Age** | | |
| ≤24 | 98 | 12.3 |
| 25–34 | 550 | 69.3 |
| ≥35 | 146 | 18.4 |
| **Religion** | | |
| Orthodox Christian | 648 | 81.6 |
| Muslim | 105 | 13.2 |
| Protestant | 32 | 4.0 |
| Others[a] | 9 | 1.1 |
| **Current marital status** | | |
| Married | 718 | 90.4 |
| Unmarried | 76 | 9.6 |
| **Women's education status** | | |
| No formal education | 96 | 12.1 |
| Primary education | 128 | 16.1 |
| Secondary education | 229 | 28.8 |
| Diploma and above | 341 | 42.9 |
| **Women's occupation** | | |
| Housewives | 357 | 45 |
| Daily laborer | 14 | 1.8 |
| Self-employee | 97 | 12.2 |
| Merchant | 97 | 12.2 |
| Government employee | 215 | 27.1 |
| **Husband educational status (n = 718)** | | |
| No formal education | 40 | 5.57 |
| Primary education | 54 | 7.52 |
| Secondary education | 149 | 20.75 |
| Diploma and above | 457 | 66.15 |
| **Husband occupation (n = 718)** | | |
| Daily labor | 45 | 6.27 |
| Self-employee | 161 | 22.42 |
| Government employee | 333 | 46.37 |
| Merchant | 151 | 21.03 |
| Others[b] | 28 | 3.89 |
| **Average family monthly income** | | |
| ≤1000 ETB | 49 | 6.2 |
| 1001–2000 ETB | 53 | 6.7 |
| ≥2001 ETB | 692 | 87.2 |

ETB: Ethiopian Birr, a: Jewish and Adventist, b: Student and Farmer.

likely to experience postpartum depression than those women who earned > 2000 ETB (AOR = 2.71, 95% CI: 1.24, 5.91). This study revealed that women who had lower decision-making power were 2.04 times more likely to have had depression during the postpartum period compared with those women who had higher decision-making power (AOR = 2.04, 95% CI: 1.31, 3.18).

**Table 2. Obstetrics, medical related and maternal health service characteristics of women who gave birth in the last one year in Gondar city, northwest Ethiopia, 2021 (n = 794).**

| Variables | Frequency | Percentage (%) |
|---|---|---|
| **Parity** | | |
| 1 | 318 | 40.1 |
| 2–4 | 446 | 56.2 |
| ≥5 | 30 | 3.8 |
| **ANC visit** | | |
| Yes | 773 | 97.4 |
| No | 21 | 2.6 |
| Number of ANC visit (n = 773) <br> <4 <br> 4 and above | <br> 274 <br> 499 | <br> 35.4 <br> 64.6 |
| **Place of delivery** | | |
| Home | 24 | 3 |
| Health facility | 770 | 97 |
| **Mode of delivery** | | |
| SVD | 471 | 59.3 |
| Cesarean section | 297 | 37.4 |
| Instrumental delivery | 26 | 3.3 |
| **PNC visit** | | |
| Yes | 418 | 52.6 |
| No | 376 | 47.4 |
| **Type of pregnancy** | | |
| Planned | 678 | 85.4 |
| Unplanned | 116 | 14.6 |
| **Was the pregnancy supported** | | |
| Yes | 737 | 92.8 |
| No | 57 | 7.2 |
| **Women's decision making power** | | |
| Higher | 494 | 62.2 |
| Lower | 300 | 37.8 |
| **Husband/partner involvement in MNCH** | | |
| Involved | 416 | 52.4 |
| Not involved | 378 | 47.6 |
| **Social support** | | |
| Poor | 228 | 28.7 |
| Moderate | 351 | 44.2 |
| Strong | 215 | 27.1 |
| **Intimate partner violence** | | |
| Yes | 388 | 48.9 |
| No | 406 | 51.1 |
| **Family history of mental health problem** | | |
| Yes | 83 | 10.5 |
| No | 711 | 89.5 |
| **Known psychiatric illness** | | |
| Yes | 19 | 2.4 |
| No | 775 | 97.6 |
| **Having information about mental health during pregnancy** | | |
| Yes | 365 | 46 |

*(Continued)*

**Table 2.** (Continued)

| Variables | Frequency | Percentage (%) |
|---|---|---|
| No | 429 | 54 |
| **Having medical illness** | | |
| Yes | 86 | 10.8 |
| No | 708 | 89.2 |
| **Experienced death of husband/friend/relatives** | | |
| Yes | 202 | 25.4 |
| No | 592 | 74.6 |

ANC: Antenatal Care, SVD: Spontaneous Vaginal Delivery, PNC: Postnatal Care, MNCH: Maternal, Neonatal and Child Health.

Women whose husband/partner were not actively involved in MNCH service were 2.34 times more likely to experience postpartum depression compared with those women whose husbands/partners were actively involved in MNCH care services (AOR = 2.34, 95%CI: 1.44, 3.81). The current study revealed that type of pregnancy has been strongly associated with postpartum depression. The odds of experiencing postpartum depression among respondents who had unplanned pregnancies were 3 times higher as compared to those women who had planned pregnancies (AOR = 3.16 95% CI: 1.77, 5.62). The study also found that there was a significant association between IPV and postpartum depression. Thus, the odds of having PPD among women who experienced IPV were about 3 times higher compared with their counterparts (AOR = 3.13; 95% CI: 1.96, 4.99) **(Table 3).**

## Discussion

This community-based cross-sectional study assessed postpartum depression and associated factors among women who gave birth in the last one year (from August 2020 to August 2021) in Gondar city, northwest Ethiopia, 2021. Thus, it was found that the prevalence of postpartum depression was 17.25% (95% CI: 14.5, 20.2), which is comparable with studies conducted in Debre Berhan, Ethiopia-15.6% [37] and Eastern Ethiopia-16.3% [38].

However, the finding of this study was higher than studies conducted in Hiwot Fana Specialized Hospital, Ethiopia-13.11% [39], Eritrea-7.4% [40], Kenya-13.0% [34], and South Africa-8.8% [41]. The possible reason for this discrepancy could be due to study setting and socio-cultural differences. All the above-mentioned studies were institution-based cross-sectional studies where women who came for MNCH care services will get health education about the physiologic and psychologic changes during the postpartum period. In addition, IPV has been linked with mental health problems, particularly postpartum depression [42]. For instance, nearly half (48.9%) of the study participants in the current study have experienced IPV, while only 3.7% of women in South Africa experienced IPV. Moreover, the possible discrepancy might be the effect of the COVID-19 pandemic and the internal conflict in the country, which may increase the prevalence of PPD in this study [43].

On the other hand, the result of this study was lower compared with other studies conducted somewhere else in Ethiopia including Gondar town-25% [44], Awi Zone-23.7% [45], Bench Maji Zone-22.4% [23], and Southwest Ethiopia-33.8% [26]. The result of this study is also lower as compared to a study conducted in Cameron-23.4% [35]. This variation might be due to the differences in the tool we used to measure the outcome variable, time of data collection, and characteristics of the study participants. The study conducted in Awi Zone, Ethiopia used Edinburgh Postnatal Depression Scale (EPDS) with a cutoff point of 8 to declare

**Table 3. Bivariable and multivariable logistic regression analysis of associated factors of postpartum depression among women who gave birth in the last 1 year in Gondar city, northwest Ethiopia, 2021 (n = 794).**

| Variables | Postpartum depression | | COR (95%CI) | AOR (95% CI) |
|---|---|---|---|---|
| | Yes | No | | |
| **Material age** | | | | |
| ≤ 24 | 31 | 67 | 3.09 (1.63, 5.88) | 2.72 (1.23, 5.85)* |
| 25–34 | 87 | 463 | 1.26 (0.74, 2.14) | 1.35 (0.73, 2.51) |
| ≥ 35 | 19 | 127 | 1 | 1 |
| **Women's education status** | | | | |
| No formal education | 30 | 66 | 3.97 (2.28, 6.93) | 1.16 (0.57, 2.39) |
| Primary education | 18 | 110 | 1.43 (0.78, 2.63) | 0.49 (0.24, 1.02) |
| Secondary education | 54 | 175 | 2.69 (1.69, 4.29) | 1.21 (0.71, 2.08) |
| Diploma and above | 35 | 306 | 1 | 1 |
| **Family monthly income** | | | | |
| ≤ 1000 ETB | 21 | 28 | 4.24 (2.32, 7.75) | 2.71 (1.24, 5.91)* |
| 1001–2000 ETB | 12 | 41 | 1.66 (0.84, 3.25) | 0.92 (0.42, 2.02) |
| ≥ 2001 ETB | 104 | 588 | 1 | 1 |
| **Place of delivery** | | | | |
| Home | 9 | 15 | 3.01 (1.29, 7.03) | 2.36 (0.89, 6,29) |
| Health facility | 128 | 642 | 1 | 1 |
| **Social support** | | | | |
| Poor | 64 | 164 | 2.97 (1.79, 4.93) | 1.53 (0.84, 2.77) |
| Moderate | 48 | 303 | 1.20 (0.72, 2.02) | 1.11 (0.62, 1.98) |
| Strong | 25 | 190 | 1 | 1 |
| **Women's decision-making power** | | | | |
| Higher | 61 | 433 | 1 | 1 |
| Lower | 76 | 224 | 2.41 (1.66, 3.49) | 2.04 (1.31, 3.18)* |
| **Husband/partner involvement in MNCH care services** | | | | |
| Involved | 43 | 373 | 1 | 1 |
| Not involved | 94 | 284 | 2.87 (1.94, 4.25) | 2.34 (1.44, 3.81)** |
| **Type pregnancy** | | | | |
| Planned | 86 | 592 | 1 | 1 |
| Unplanned | 51 | 65 | 5.40 (3.51, 8.31) | 3.16 (1.77, 5.62)** |
| **Was the recent pregnancy supported** | | | | |
| Yes | 118 | 619 | 1 | 1 |
| No | 19 | 38 | 2.62 (1.46, 2.71) | 0.53 (0.23, 1.19) |
| **Intimate partner violence** | | | | |
| Yes | 104 | 284 | 4.41 (2.71, 6.31) | 3.13 (1.96, 4.99)** |
| No | 33 | 373 | 1 | 1 |

NB

* Significant (P-value ≤ 0.05)

** P-value <0.001.

MNCH: Maternal, Neonatal, and Child Health, AOR: Adjusted Odd Ratio, COR: Crude Odd Ratio, CI: Confidence Interval.

postpartum depression. In this study, however, the PHQ-9 depression scale with a cutoff point of 10 was used. Moreover, the study participants in Gondar town and Cameron were women who gave birth in the last 6 weeks and women whose children aged 4 to 6 weeks, respectively. As a result, the prevalence of PPD might be increased because postpartum depression is most common in the first 6 weeks after child birth [23]. On the other hand, the study in Bench Maji Zone, Ethiopia showed that 42.1% of the study participants were under the age of 23 years. However, only 11.2% of the participants were under the age of 24 in this study. It has been evidenced that being younger age is highly correlated with postpartum depression [32,46]. The lower prevalence of PPD in the current study might also be related to the low incidence of unplanned pregnancy. Unplanned and unsupported pregnancies have been associated with PPD as evidenced by scholars [22,23,47,48]. Hence, 30% of the pregnancy in the Southwest Ethiopia study was unplanned whereas only 14.6% of the pregnancies were unplanned in our study.

It has been found that being younger age increases the odds of developing PPD. Accordingly, the odds of having postpartum depression was 2.72 times higher among women aged ≤ 24 years compared with those women aged ≥ 35 years old. This finding was supported by the study conducted in Southwest Ethiopia [23], Sudan [33], Kenya [49], and Armenia [32]. The possible explanation might be due to the fact that younger women are expected to be exposed to emotional distress as they experience childbirth for the first time. Besides, the additional burden of caring for infants and preparation to be a mother will be often challenging and will lead to unpleasant health outcomes [50]. In this regard, the need to screen younger women for mental health problems and endorsing screening tools starting from the prenatal period will be crucial.

The study also affirmed that the family monthly income of the respondents was one of the variables positively associated with PPD. Respondents who had an average monthly income ≤ 1000 ETB were 2.7 times more likely to report PPD than participants who had an average monthly income of >2000 ETB. This finding was supported by studies conducted in Kenya [34] and Cameron [35]. This might be due to women with low socioeconomic status may face difficulties to fulfill their needs and newborns during the postpartum period. Low socioeconomic status has been connected with a high rate of different mental health problems [51,52].

This study also revealed that women who had lower decision-making power were 2.04 times more likely to have had depression during the postpartum period compared with those women who had higher decision-making power. This result is consistent with studies done in Bahir Dar, Ethiopia [21], and China [36]. The possible reason for this could be those women who didn't have power and control over resources, restriction to access maternal and reproductive health services, and unable to decide independently for their health could negatively affect their overall wellbeing. This could also be justifies as about 45% of the participants in this study were housewives, in which being unemployed has been associated with mental health problems so far [53]. This is because employed women are expected to have higher levels of household decision-making power as compared with their counterparts. Unemployed women are usually economically dependent on their husband/partner, particularly in developing countries and exclude women from different opportunities [54].

In this study, husband/partner involvement in MNCH service was significantly associated with PPD. Accordingly, women whose husbands/partners were not actively involved in MNCH care service were 2.34 times more likely to experience postpartum depression compared with those women whose husbands/partners were involved in MNCH care service. This could be explained by having husband/partner involvement in MNCH care services may build a higher sense of support for the women. The other possible justification could be that men are

influential in health care decision-making, which leads to a woman's experiencing mental health problems. Evidence support that husband/partner involvement in MNCH services is found to be crucial for the reduction of adverse health outcomes [55]. However, only 52.4% of husbands/partners have been involved in MNCH care service in the current study.

The current study revealed that the type of pregnancy has been strongly associated with PPD. Thus, the odds of experiencing PPD among respondents who had unplanned pregnancy was 3 times more compared with those women who had planned pregnancy. This result is supported by studies conducted in Ethiopia such as Bahir Dar [21], Nekemte town [22], Bench Maji zone, [23], and southwest Ethiopia [26], in Kenya [49], in Nepal [56], and in Pennsylvania [57]. This could be due to the fact that pregnancy itself needs physiological, psychological, and financial preparation. Empirical evidence showed that unplanned pregnancy is associated with PPD [47,48].

This study also declared that there was a significant association between IPV and PPD. Hence, the odds of having PPD among women who experienced IPV were 3 times higher compared with their counterparts. Similar findings were reported from previous studies done in Ethiopia [14,21,22], Nigeria [58], Bangladesh [8], and Canada [59]. The explanation for this might be IPV has a major effect on women's physical and psychological health and this may lead to postpartum depression [9]. This indicates that screening for IPV in antenatal and postnatal care could help to identify and treat women at risk of depression.

We authors strongly believe that the present study is very important in providing evidences about the prevalence and its associated factors of PPD. Based on this evidence, policymakers should think about the burden of PPD, which is left undiagnosed and untreated due to the lack of an integrated depression screening tool with prenatal and postnatal care and low PNC service utilization. Lastly, the authors would like to acknowledge the limitation of this study. Due to the cross-sectional nature of the study, it couldn't be possible to infer cause-effect between the outcome and explanatory variables. Our study did not assess the effect of the COVID-19 pandemic and internal conflict-related issue. The use of interviewer-administer questions can lead to social desirability bias. This might cause to underestimate of the prevalence of PPD. Moreover, since we include women who gave birth in the last year recall bias might be expected.

## Conclusion

In this study, PPD was comparable with previous studies. However, given the community health context, it is an important public health issue. Being younger, lack of husband involvement in MNCH care services, lack of decision-making power in the household, experiencing IPV, unplanned pregnancy, and lower household monthly income were increase the odds of experiencing PPD. It is important to integrate routine screening and management tools for PPD with prenatal and PNC service guidelines for screening, timely transfer, and early treatment of those women who are at risk of postpartum depression, so as to improve maternal and child wellbeing in general. Therefore, the government and non-governmental organizations should focus on this public health problem because PPD has a potential adverse effect on parenting practices and children's physical and emotional development [34,60]. It is also crucial to advocate the need for the husband's/partner involvement in MNCH services and ensure women's decision-making power in the household. Moreover, community-based sexual and reproductive health education would be better to reduce risk factors for postpartum depression.

Healthcare providers who work directly with pregnant and postpartum women have a better opportunity to identify the risks, signs, and symptoms of PPD and refer patients for

treatment. They also focus on preconception care for the prevention of unplanned pregnancy and provide psychotherapy for IPV victims for the prevention of PPD. The healthcare provider prefers to give counseling about the impact of husband's/partner involvement in maternal and child health.

For future researchers, we recommend an advanced study design that would strongly infer the casual link between exposure to varying forms of independent factors and the development of PPD. In addition, we recommend a qualitative research to explore PPD in depth.

## Supporting information

**S1 File. English version of the questionnaire.**
(DOCX)

**S2 File. SPSS dataset.**
(SAV)

## Acknowledgments

We would like to thank the University of Gondar for providing study ethical clearance to conduct this study. Our gratitude also goes to all data collectors and study participants. We are glad to Gondar city kebeles for writing permission letter.

## Author Contributions

**Conceptualization:** Azmeraw Ambachew Kebede.

**Data curation:** Azmeraw Ambachew Kebede, Dereje Nibret Gessesse, Mastewal Belayneh Aklil, Wubedle Zelalem Temesgan, Marta Yimam Abegaz, Tazeb Alemu Anteneh, Nebiyu Solomon Tibebu, Haymanot Nigatu Alemu, Tsion Tadesse Haile, Asmra Tesfahun Seyoum, Agumas Eskezia Tiguh, Ayenew Engida Yismaw, Muhabaw Shumye Mihret, Goshu Nenko, Kindu Yinges Wondie, Birhan Tsegaw Taye, Nuhamin Tesfa Tsega.

**Formal analysis:** Azmeraw Ambachew Kebede, Dereje Nibret Gessesse, Mastewal Belayneh Aklil, Wubedle Zelalem Temesgan, Marta Yimam Abegaz, Tazeb Alemu Anteneh, Nebiyu Solomon Tibebu, Haymanot Nigatu Alemu, Tsion Tadesse Haile, Asmra Tesfahun Seyoum, Agumas Eskezia Tiguh, Ayenew Engida Yismaw, Muhabaw Shumye Mihret, Goshu Nenko, Kindu Yinges Wondie, Birhan Tsegaw Taye, Nuhamin Tesfa Tsega.

**Funding acquisition:** Azmeraw Ambachew Kebede, Dereje Nibret Gessesse, Mastewal Belayneh Aklil, Wubedle Zelalem Temesgan, Marta Yimam Abegaz, Tazeb Alemu Anteneh, Nebiyu Solomon Tibebu, Haymanot Nigatu Alemu, Tsion Tadesse Haile, Asmra Tesfahun Seyoum, Agumas Eskezia Tiguh, Ayenew Engida Yismaw, Muhabaw Shumye Mihret, Goshu Nenko, Kindu Yinges Wondie, Birhan Tsegaw Taye, Nuhamin Tesfa Tsega.

**Investigation:** Azmeraw Ambachew Kebede, Dereje Nibret Gessesse, Mastewal Belayneh Aklil, Wubedle Zelalem Temesgan, Marta Yimam Abegaz, Tazeb Alemu Anteneh, Nebiyu Solomon Tibebu, Haymanot Nigatu Alemu, Tsion Tadesse Haile, Asmra Tesfahun Seyoum, Agumas Eskezia Tiguh, Ayenew Engida Yismaw, Muhabaw Shumye Mihret, Goshu Nenko, Kindu Yinges Wondie, Birhan Tsegaw Taye, Nuhamin Tesfa Tsega.

**Methodology:** Azmeraw Ambachew Kebede, Dereje Nibret Gessesse, Mastewal Belayneh Aklil, Wubedle Zelalem Temesgan, Marta Yimam Abegaz, Tazeb Alemu Anteneh, Nebiyu Solomon Tibebu, Haymanot Nigatu Alemu, Tsion Tadesse Haile, Asmra Tesfahun Seyoum,

Agumas Eskezia Tiguh, Ayenew Engida Yismaw, Muhabaw Shumye Mihret, Goshu Nenko, Kindu Yinges Wondie, Birhan Tsegaw Taye, Nuhamin Tesfa Tsega.

**Project administration:** Azmeraw Ambachew Kebede, Dereje Nibret Gessesse, Mastewal Belayneh Aklil, Wubedle Zelalem Temesgan, Marta Yimam Abegaz, Tazeb Alemu Anteneh, Nebiyu Solomon Tibebu, Haymanot Nigatu Alemu, Tsion Tadesse Haile, Asmra Tesfahun Seyoum, Agumas Eskezia Tiguh, Ayenew Engida Yismaw, Muhabaw Shumye Mihret, Goshu Nenko, Kindu Yinges Wondie, Birhan Tsegaw Taye, Nuhamin Tesfa Tsega.

**Resources:** Azmeraw Ambachew Kebede, Dereje Nibret Gessesse, Mastewal Belayneh Aklil, Wubedle Zelalem Temesgan, Marta Yimam Abegaz, Tazeb Alemu Anteneh, Nebiyu Solomon Tibebu, Haymanot Nigatu Alemu, Tsion Tadesse Haile, Asmra Tesfahun Seyoum, Agumas Eskezia Tiguh, Ayenew Engida Yismaw, Muhabaw Shumye Mihret, Goshu Nenko, Kindu Yinges Wondie, Birhan Tsegaw Taye, Nuhamin Tesfa Tsega.

**Software:** Azmeraw Ambachew Kebede, Dereje Nibret Gessesse, Mastewal Belayneh Aklil, Wubedle Zelalem Temesgan, Marta Yimam Abegaz, Tazeb Alemu Anteneh, Nebiyu Solomon Tibebu, Haymanot Nigatu Alemu, Tsion Tadesse Haile, Asmra Tesfahun Seyoum, Agumas Eskezia Tiguh, Ayenew Engida Yismaw, Muhabaw Shumye Mihret, Goshu Nenko, Kindu Yinges Wondie, Birhan Tsegaw Taye, Nuhamin Tesfa Tsega.

**Supervision:** Azmeraw Ambachew Kebede, Dereje Nibret Gessesse, Mastewal Belayneh Aklil, Wubedle Zelalem Temesgan, Marta Yimam Abegaz, Tazeb Alemu Anteneh, Nebiyu Solomon Tibebu, Haymanot Nigatu Alemu, Tsion Tadesse Haile, Asmra Tesfahun Seyoum, Agumas Eskezia Tiguh, Ayenew Engida Yismaw, Muhabaw Shumye Mihret, Goshu Nenko, Kindu Yinges Wondie, Birhan Tsegaw Taye, Nuhamin Tesfa Tsega.

**Validation:** Azmeraw Ambachew Kebede, Dereje Nibret Gessesse, Mastewal Belayneh Aklil, Wubedle Zelalem Temesgan, Marta Yimam Abegaz, Tazeb Alemu Anteneh, Nebiyu Solomon Tibebu, Haymanot Nigatu Alemu, Tsion Tadesse Haile, Asmra Tesfahun Seyoum, Agumas Eskezia Tiguh, Ayenew Engida Yismaw, Muhabaw Shumye Mihret, Goshu Nenko, Kindu Yinges Wondie, Birhan Tsegaw Taye, Nuhamin Tesfa Tsega.

**Visualization:** Azmeraw Ambachew Kebede, Dereje Nibret Gessesse, Mastewal Belayneh Aklil, Wubedle Zelalem Temesgan, Marta Yimam Abegaz, Tazeb Alemu Anteneh, Nebiyu Solomon Tibebu, Haymanot Nigatu Alemu, Tsion Tadesse Haile, Asmra Tesfahun Seyoum, Agumas Eskezia Tiguh, Ayenew Engida Yismaw, Muhabaw Shumye Mihret, Goshu Nenko, Kindu Yinges Wondie, Birhan Tsegaw Taye, Nuhamin Tesfa Tsega.

**Writing – original draft:** Azmeraw Ambachew Kebede, Wubedle Zelalem Temesgan, Tazeb Alemu Anteneh, Nuhamin Tesfa Tsega.

**Writing – review & editing:** Azmeraw Ambachew Kebede, Dereje Nibret Gessesse, Mastewal Belayneh Aklil, Wubedle Zelalem Temesgan, Marta Yimam Abegaz, Tazeb Alemu Anteneh, Nebiyu Solomon Tibebu, Haymanot Nigatu Alemu, Tsion Tadesse Haile, Asmra Tesfahun Seyoum, Agumas Eskezia Tiguh, Ayenew Engida Yismaw, Muhabaw Shumye Mihret, Goshu Nenko, Kindu Yinges Wondie, Birhan Tsegaw Taye, Nuhamin Tesfa Tsega.

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
