## [Decision Letter · Decision Letter 0]

21 Apr 2022

PONE-D-21-35177

Low husband involvement in maternal and child health services and intimate partner violence increases the odds of postpartum depression in northwest Ethiopia: a community-based study

PLOS ONE

Dear Dr. Tsega,

Thank you for submitting your manuscript to PLOS ONE. After careful consideration, we feel that it has merit but does not fully meet PLOS ONE’s publication criteria as it currently stands. Therefore, we invite you to submit a revised version of the manuscript that addresses the points raised during the review process.

We look forward to receiving your revised manuscript.

Kind regards,

Ammal Mokhtar Metwally, Ph.D (MD)

Academic Editor

PLOS ONE

Additional Editor Comments:

Please note that your manuscript was reviewed by 2 experts in the field. There is consensus agreement that the idea of the article is interesting. Meanwhile, some of the reviewers identified important problems in your submission and provided copious comments. Please consider responding to the reviewers’ remarks. The manuscript could be greatly strengthened by considering editing according to the specific mentioned comments.

Reviewers' comments:

Reviewer's Responses to Questions

**Comments to the Author**

1. Is the manuscript technically sound, and do the data support the conclusions?

Reviewer #1: Yes

Reviewer #2: Yes

2. Has the statistical analysis been performed appropriately and rigorously? 

Reviewer #1: Yes

Reviewer #2: I Don't Know

3. Have the authors made all data underlying the findings in their manuscript fully available?

Reviewer #1: Yes

Reviewer #2: Yes

4. Is the manuscript presented in an intelligible fashion and written in standard English?

Reviewer #1: Yes

Reviewer #2: Yes

5. Review Comments to the Author

Reviewer #1: There are a number of terms that need additional explanation or definitions throughout the text. There are statements that require more clarity. The discussion section doesn't really tell us anything. Such a huge part of it is spent discussing the differences in PPD prevalence between your study and other studies, which isn't really telling the reader anything helpful. Very little time is spent talking about what you learned from the study that is new information. There is no discussion of future research that is needed. There are no specific recommendations about what to do to address the issues addressed in your paper. I am left think so what when I read the paper. Yes we now know this about PPD in Gondor, but what should be done to address that issue? What else should be considered when looking more into this topic in that setting or in other parts of Ethiopia? The discussion and conclusion sections need to be reworked so they are actually telling us something useful.

Reviewer #2: Overall, the manuscript is good and has been organized well. I have a few comments to improve the manuscript as follows:

1. Abstract: the abstract is good and concise. However, I noticed you didn’t write anything about the discussion that you mentioned in the manuscript.

-in the result of the abstract section, you mentioned MNCH and in the conclusion, you have written, ‘maternal and child health’. Two are the same things. please keep consistency.

2. Introduction: You could add some references on IPV in the national and local context that could strengthen your argument in the paper.

- you mentioned you did not compare the PPD screening tool with the national guideline? Why? What was the gap in the national guideline? Did you explore or analyze it?

3. Method: In the method section, you said that you choose samples from mothers who were first- time mothers and who had experienced multiple pregnancies. Can you mention how many Primi mothers you have interviewed who were suffering from depression?

-can you explain clearly the lottery method or any reference?

-Two data collectors completed 757 interviews while nonresponse rate was 10%. They collected data within a single month. How many interviews did they conduct per day? How the data quality was checked? Can you explain it a bit more?

4. Result:

You have mentioned women’s socio-cultural aspects were linked to postpartum depression. How is religious background linked to depression? Would you please explain?

- 92.8% of pregnancy was planned and had family support during pregnancy that you mentioned. If so how this variable was selected as the cause of postpartum depression?

- Family income is another factor you mentioned. I am saying, poverty is a common reason for mental stress in any situation. Why did you particularly mention it as a cause of postpartum depression? Please explain.

- You have focused on IPV more than any other variables you mentioned in the manuscript. But IPV itself is a big area that leads to worse mental health situations. I think you did not mention all the component of IPV that leads to postpartum depression. I encourage you to recheck your data and dig down the path.

5. Discussion: you have written well, but it would be stronger if you could add more references on these issues you have explored in your research.

-What about the national data about postpartum depression. Does it support your data? Did you check other studies in Ethiopia’s other regions that support your data?

-What about the policy implication? How did your data recommend to improve the mental health situation of the women in the country? Please explain

6. PLOS authors have the option to publish the peer review history of their article (what does this mean?). If published, this will include your full peer review and any attached files.

Reviewer #1: No

Reviewer #2: No

---

## [Decision Letter · Decision Letter 1]

19 Jul 2022

PONE-D-21-35177R1Low husband involvement in maternal and child health services and intimate partner violence increases the odds of postpartum depression in northwest Ethiopia: a community-based studyPLOS ONE

Dear Dr. Tsega,

Thank you for submitting your manuscript to PLOS ONE. After careful consideration, we feel that it has merit but does not fully meet PLOS ONE’s publication criteria as it currently stands. Therefore, we invite you to submit a revised version of the manuscript that addresses the points raised during the review process.

We look forward to receiving your revised manuscript.

Kind regards,

Ammal Mokhtar Metwally, Ph.D (MD)

Academic Editor

PLOS ONE

Journal Requirements:

Additional Editor Comments:

Great effort was made by the authors to utilize the feedback that was provided for them to correct their manuscript. I find it interesting and improved with respect to the original submission. Please consider responding to the reviewers’ remarks. The manuscript could be greatly strengthened by considering editing according to the specific mentioned comments.

Reviewers' comments:

Reviewer's Responses to Questions

**Comments to the Author**

1. If the authors have adequately addressed your comments raised in a previous round of review and you feel that this manuscript is now acceptable for publication, you may indicate that here to bypass the “Comments to the Author” section, enter your conflict of interest statement in the “Confidential to Editor” section, and submit your "Accept" recommendation.

Reviewer #1: (No Response)

Reviewer #2: All comments have been addressed

2. Is the manuscript technically sound, and do the data support the conclusions?

Reviewer #1: Yes

Reviewer #2: Yes

3. Has the statistical analysis been performed appropriately and rigorously? 

Reviewer #1: Yes

Reviewer #2: I Don't Know

4. Have the authors made all data underlying the findings in their manuscript fully available?

Reviewer #1: Yes

Reviewer #2: Yes

5. Is the manuscript presented in an intelligible fashion and written in standard English?

Reviewer #1: Yes

Reviewer #2: Yes

6. Review Comments to the Author

Reviewer #1: Because of the timing of this study, it is important to acknowledge the potential impacts of the COVID 19 pandemic and the conflict in Tigray, as it isn’t too far from Gondor City, on the mental health of the women included in the study.

Evidence collected around the world has shown an increase in IPV during the pandemic, and this would likely have an impact on PPD. Also the pandemic itself has had a massive effect on mental health directly. There is also evidence that living in or near a conflict zone has dramatic impacts on mental health, further exacerbating the chances of PPD. Your study didn’t collect data related to this, but at least acknowledging the potential impacts and that you are comparing your findings to studies pre-COVID could skew the data.

You don’t really “have” IPV, it isn’t a disease, it is something that you experience or is done to you. The language around this topic should be adjusted.

There still needs to be more information in the conclusion/discussion about what the outcomes from this study should be. You talk about policymakers and researchers a little but, but what about practitioners who will be working directly with these patients? What do you recommend as to screening? I'm still not clear on what your recommendations are regarding regular screening for PPD.

Reviewer #2: Review report of D-21-35177:

I would like to thank to the authors who addressed majority of the comments on the issues I raised during my first review. This is an important manuscript having data with mental health issues.

However, still I didn’t understand the lottery method theoretically. If possible, explain it or provide a reference which will help readers to understand.

Wish you good luck!

7. PLOS authors have the option to publish the peer review history of their article (what does this mean?). If published, this will include your full peer review and any attached files.

Reviewer #1: No

Reviewer #2: No

---

## [Author Response · Author response to Decision Letter 1]

1 Sep 2022

Date: August 31/ 2022 

Point by point response to reviewers comment 

Manuscript title: Low husband involvement in maternal and child health services and intimate partner violence increases the odds of postpartum depression in northwest Ethiopia: a community-based study.

Manuscript Ref: Submission ID PONE-D-21-35177R1

We are very grateful to both the editor and reviewers for your comments and concerns for the betterment of our manuscript. Appreciating your effort and valuable comments, we have provided possible reflections on the raised concerns and questions. Kindly find our responses here. In addition, we incorporated your comments and suggestions in the revised manuscript. 

Response to reviewer’s comments 

Reviewer Comments:

#Reviewer 1

1. Because of the timing of this study, it is important to acknowledge the potential impacts of the COVID 19 pandemic and the conflict in Tigray, as it isn’t too far from Gondar City, on the mental health of the women included in the study. Evidence collected around the world has shown an increase in IPV during the pandemic, and this would likely have an impact on PPD. Also the pandemic itself has had a massive effect on mental health directly. There is also evidence that living in or near a conflict zone has dramatic impacts on mental health, further exacerbating the chances of PPD. Your study didn’t collect data related to this, but at least acknowledging the potential impacts and that you are comparing your findings to studies pre-COVID could skew the data. 

Author’s response: Thank you dear reviewer for your insightful comment. It has been considered in the revised manuscript. 

2. You don’t really “have” IPV, it isn’t a disease, and it is something that you experience or is done to you. The language around this topic should be adjusted. 

Author’s response: Dear reviewer, thank you for your important comment. It has been corrected in the revised manuscript. Please look at the revised manuscript. 

3. There still needs to be more information in the conclusion/discussion about what the outcomes from this study should be. You talk about policymakers and researchers a little, but what about practitioners who will be working directly with these patients? What do you recommend as to screening? I'm still not clear on what your recommendations are regarding regular screening for PPD.

Author’s response: Thank you dear reviewer for your important comment. We have tried to add further recommendations. Please look at the revised manuscript. 

#Reviewer 2

1. I would like to thank to the authors who addressed majority of the comments on the issues I raised during my first review. This is an important manuscript having data with mental health issues. However, still I didn’t understand the lottery method theoretically. If possible, explain it or provide a reference which will help readers to understand.

Author’s response: Dear reviewer, thank you for your invaluable contribution to the improvement of our manuscript. We simply meant that simple random sampling method. The lottery method is one type of simple random sampling technique, and we used it by assigning a number to each kebele (the smallest administrative unit), after which numbers are selected at random.

---

## [Decision Letter · Decision Letter 2]

14 Oct 2022

Low husband involvement in maternal and child health services and intimate partner violence increases the odds of postpartum depression in northwest Ethiopia: a community-based study

PONE-D-21-35177R2

Dear Dr. Tsega,

We’re pleased to inform you that your manuscript has been judged scientifically suitable for publication and will be formally accepted for publication once it meets all outstanding technical requirements.

Kind regards,

Ammal Mokhtar Metwally, Ph.D (MD)

Academic Editor

PLOS ONE

Additional Editor Comments (optional):

Reviewers' comments:

Reviewer's Responses to Questions

**Comments to the Author**

1. If the authors have adequately addressed your comments raised in a previous round of review and you feel that this manuscript is now acceptable for publication, you may indicate that here to bypass the “Comments to the Author” section, enter your conflict of interest statement in the “Confidential to Editor” section, and submit your "Accept" recommendation.

Reviewer #1: All comments have been addressed

Reviewer #2: All comments have been addressed

2. Is the manuscript technically sound, and do the data support the conclusions?

Reviewer #1: (No Response)

Reviewer #2: Yes

3. Has the statistical analysis been performed appropriately and rigorously? 

Reviewer #1: (No Response)

Reviewer #2: I Don't Know

4. Have the authors made all data underlying the findings in their manuscript fully available?

Reviewer #1: (No Response)

Reviewer #2: Yes

5. Is the manuscript presented in an intelligible fashion and written in standard English?

Reviewer #1: (No Response)

Reviewer #2: Yes

6. Review Comments to the Author

Reviewer #1: (No Response)

Reviewer #2: This manuscript have met its full criteria. I think this will contribute a lot to public health researcher for future research.

7. PLOS authors have the option to publish the peer review history of their article (what does this mean?). If published, this will include your full peer review and any attached files.

Reviewer #1: No

Reviewer #2: No

---

## [Editor Report · Acceptance letter]

17 Oct 2022

PONE-D-21-35177R2 

Low husband involvement in maternal and child health services and intimate partner violence increases the odds of postpartum depression in northwest Ethiopia: a community-based study 

Dear Dr. Tsega:

I'm pleased to inform you that your manuscript has been deemed suitable for publication in PLOS ONE. Congratulations! Your manuscript is now with our production department. 

Kind regards, 

on behalf of

Professor Ammal Mokhtar Metwally 

Academic Editor

PLOS ONE